# Convergent evolution of plant pattern recognition receptors sensing cysteine-rich patterns from three microbial kingdoms

Yuankun Yang [1,10] ✉, Christina E. Steidele[1,2,10], Clemens Rössner [3], Birgit Löffelhardt[1], Dagmar Kolb[1], Thomas Leisen[4], Weiguo Zhang[1,5], Christina Ludwig [6], Georg Felix[1], Michael F. Seidl [7,8], Annette Becker [3], Thorsten Nürnberger [1], Matthias Hahn[4], Bertolt Gust[9], Harald Gross [9], Ralph Hückelhoven[2] & Andrea A. Gust [1] ✉

The *Arabidopsis thaliana* Receptor-Like Protein RLP30 contributes to immunity against the fungal pathogen *Sclerotinia sclerotiorum*. Here we identify the RLP30-ligand as a small cysteine-rich protein (SCP) that occurs in many fungi and oomycetes and is also recognized by the *Nicotiana benthamiana* RLP RE02. However, RLP30 and RE02 share little sequence similarity and respond to different parts of the native/folded protein. Moreover, some Brassicaceae other than *Arabidopsis* also respond to a linear SCP peptide instead of the folded protein, suggesting that SCP is an eminent immune target that led to the convergent evolution of distinct immune receptors in plants. Surprisingly, RLP30 shows a second ligand specificity for a SCP-nonhomologous protein secreted by bacterial Pseudomonads. RLP30 expression in *N. tabacum* results in quantitatively lower susceptibility to bacterial, fungal and oomycete pathogens, thus demonstrating that detection of immunogenic patterns by *Arabidopsis* RLP30 is involved in defense against pathogens from three microbial kingdoms.

To survive attacks by harmful microbes, plants have evolved a multifaceted innate immune system[1–3]. Pattern recognition receptors (PRRs) that perceive pathogen/microbe-associated molecular patterns (PAMPs/MAMPs) at the plasma membrane play an important role in this system to initiate active defense responses[3–6]. Many PRRs contain leucine-rich repeat (LRR) ectodomains. Based on the presence or absence of a cytoplasmic kinase domain, these LRR-PRRs can be classified as receptor kinases (LRR-RKs) or receptor-like proteins (LRR-RPs), respectively[3–6]. Immunity-related LRR-RPs form constitutive complexes with the common adapter kinase SOBIR1 (SUPPRESSOR OF BRASSINOSTEROID INSENSITIVE 1 (BRI1)-ASSO-CIATED KINASE (BAK1)-INTERACTING RECEPTOR KINASE 1), thus forming bi-molecular equivalents of LRR-RKs[4,7]. Upon binding their specific PAMP ligands, the LRR-RP/SOBIR1 complexes, much like the LRR-RKs, recruit co-receptors of the SERK (SOMATIC EMBRY-OGENESIS RECEPTOR KINASE) family such as BAK1 for activation of

[1]Department of Plant Biochemistry, Center of Plant Molecular Biology (ZMBP), Eberhard-Karls-University of Tübingen, Tübingen, Germany. [2]Chair of Phyto-pathology, TUM School of Life Sciences, Technische Universität München, Freising-Weihenstephan, Germany. [3]Institute of Botany, Developmental Biology of Plants, Justus-Liebig-University Gießen, Gießen, Germany. [4]Department of Biology, Phytopathology group, Technical University of Kaiserslautern, Kaiserslautern, Germany. [5]Faculty of Life Science, Northwest University, Xi'an, China. [6]Bavarian Center for Biomolecular Mass Spectrometry, TUM School of Life Sciences, Technische Universität München, Freising-Weihenstephan, Germany. [7]Theoretical Biology & Bioinformatics, Department of Biology, Utrecht University, Utrecht, The Netherlands. [8]Laboratory of Phytopathology, Wageningen University & Research, Wageningen, The Netherlands. [9]Department of Pharmaceutical Biology, Pharmaceutical Institute, Eberhard-Karls-University of Tübingen, Tübingen, Germany. [10]These authors contributed equally: Yuankun Yang, Christina E. Steidele. ✉e-mail: yuankun.yang@zmbp.uni-tuebingen.de; andrea.gust@zmbp.uni-tuebingen.de

intracellular immune signaling and induction of defense responses[4,8].

Over the past decades, a number of PAMP/PRR pairs have been identified[3,4,6,9] and most of these PAMPs occur only in a single kingdom of microorganisms. However, there are notable exceptions including the XEG1-type glucanohydrolases and the VmE02-type cysteine-rich proteins which are conserved in many plant-pathogenic fungi and oomycetes, or NLPs (NEP1-(NECROSIS AND ETHYLENE-INDUCING PEPTIDE 1)-LIKE PROTEINS) which are present in bacterial, fungal and oomycete species[10–13].

Perception of PAMPs is often restricted to closely related species, reflecting a vast diversity in the repertoires of PRRs between plant families. Restriction to closely related species holds for LRR-RP-type PRRs of *Arabidopsis*, including RLP23, RLP32 and RLP42, which sense nlp20, proteobacterial TRANSLATION INITIATION FACTOR 1 (IF1) and fungal polygalacturonases (PGs), respectively[4,14,15]. These LRR-RPs have no obvious homologs in species outside the Brassicaceae family. Likewise, LRR-RPs identified in solanaceous plants such as tomato, including EIX2 (ETHYLENE-INDUCING XYLANASE RECEPTOR 2) and the Cf proteins detecting effectors from *Cladosporium fulvum* have no counterparts in species of other plant families[4]. Intriguingly, detection of only a few PAMPs, notably bacterial flagellin by FLAGELLIN-SENSING 2 (FLS2) and fungal chitin by CHITIN-ELICITOR RECEPTOR KINASE 1 (CERK1), occurs in a broad range of plant species[16,17].

The *Arabidopsis* Receptor-Like Protein 30 (RLP30) mediates basal resistance to the fungus *Sclerotinia sclerotiorum* but the identity of the proteinaceous PAMP detected by RLP30 remained open[18]. Here, we report the identification of this factor as a SMALL CYSTEINE-RICH PROTEIN (SCP) closely related to VmE02 from *Valsa mali*[12]. Apart from sensing SCPs from different fungi and oomycetes, RLP30 can recognize an SCP-unrelated pattern from *Pseudomonas* bacteria and confers increased resistance to *Nicotiana tabacum* plants. We further reveal distinct immune-sensing mechanisms for SCPs in Brassicaceae and Solanaceae species. Our findings unveil an unanticipated complex relationship between PRRs and PAMPs where a single PRR recognizes PAMPs from microorganisms of three kingdoms and where a single PAMP is detected by phylogenetically distinct PRRs from different plant species.

## Results

### SCP$^{Ss}$ is the immunogenic pattern secreted by *S. sclerotiorum*

The *Arabidopsis* RLP30 detects a proteinaceous elicitor secreted by *S. sclerotiorum*, originally termed SCFE1[18]. To identify the immunogenic pattern in the supernatants of fungal cultures, we used an extended fractionation protocol and monitored fractions for RLP30-dependent induction of the plant stress hormone ethylene (Supplementary Fig. 1a, b). Mass spectrometric analysis of the trypsin fragments obtained with the ethylene-inducing fractions revealed the presence of a small cysteine-rich protein (Sscle06g048920; SMALL CYSTEINE-RICH PROTEIN FROM *S. SCLEROTIORUM*, SCP$^{Ss}$) of 147 amino acids containing eight cysteine residues but no known protein domains (Supplementary Table 1 and Supplementary Fig. 1c). To confirm SCP$^{Ss}$ as the ligand of RLP30, the SCP$^{Ss}$ protein was heterologously expressed in *N. benthamiana* or *Pichia pastoris* (Supplementary Fig. 1d, e), with C-terminal GFP or His tags, respectively. Both forms were found to induce ethylene production in *Arabidopsis* Col-0 wild-type plants but not in the *rlp30-2* mutant (Fig. 1a and Supplementary Fig. 1f), ruling out that contaminations occurring during the purification procedure or SCP$^{Ss}$-copurifying proteins are responsible for the RLP30-dependent defense response. His-tagged SCP$^{Ss}$, affinity purified via His-Trap columns, triggered ethylene production in Col-0 plants at nanomolar concentrations with a half-maximal induction (EC$_{50}$) at ~26 nM (Supplementary Fig. 2a). In contrast to *rlp30-2* mutants, Col-0 wild-type plants and *rlp30-2* mutants complemented with RLP30 responded to

the SCP$^{Ss}$ preparations with typical PAMP responses such as induction of reactive oxygen species (ROS), activation of mitogen-activated protein kinases (MAPK), and a generally induced state of resistance, supported by significantly reduced colonization by the fungus *Botrytis cinerea* or the bacterial pathogen *Pseudomonas syringae* pv. *tomato* (*Pst*) DC3000 (Fig. 1b, c and Supplementary Fig. 2b, c). Unlike most other known patterns recognized by RPs, SCP$^{Ss}$ did not only elicit ethylene production in *Arabidopsis* and related *Brassica* species, but also in solanaceous plants such as *N. benthamiana*, tomato, pepper, and potato (Fig. 1d).

To assess whether SCP$^{Ss}$ can be found in complex with RLP30, co-immunoprecipitation assays were performed with extracts from *N. benthamiana* leaves that were transiently co-transformed with myc-tagged SCP$^{Ss}$ and with either GFP-tagged RLP30 or the GFP-tagged nlp20 receptor RLP23[19] as control. SCP$^{Ss}$-myc co-precipitated specifically with RLP30 but not with the structurally related RLP23 (Fig. 1e).

Several *Arabidopsis* accessions that were found to exhibit no response to *S. sclerotiorum* extracts carry single nucleotide polymorphisms in their *RLP30* genes leading to single amino acid changes[18]. In an extended screen with 77 accessions, we identified further accessions that did not respond to SCP$^{Ss}$ and that carried point mutations in *RLP30* (Supplementary Figs. 3a and 4). We tested the altered RLP30 versions occurring in these insensitive accessions in a co-precipitation assay for binding of SCP$^{Ss}$. While most receptor versions, except RLP30 from Sq-1 and Lerik1-3, were expressed in *N. benthamiana*, none of them precipitated SCP$^{Ss}$ (Supplementary Fig. 3b). Thus, the single amino acid substitutions in different parts of the LRR and island domains, which are specific to SCP$^{Ss}$-responsive RLP30 variants[18], such as L307R in Mt-0, R433G in Lov-1 and Lov-5, R433L in ICE111, N561Y in Bak-2, G563V in Br-0, and F760 in Gu-0 (Supplementary Fig. 4), respectively, all affect RLP30/SCP$^{Ss}$ complex formation and, likely, are responsible for the non-functionality of these mutated receptors.

### RLP30 detects SCPs from fungi and oomycetes

Homologs of SCP$^{Ss}$ exist in many fungi and oomycetes[12]. They typically contain a predicted signal peptide, seven conserved motifs (R1 to R7) and eight conserved cysteine residues (Supplementary Fig. 5a), however, the absence of previously characterized protein domains hampers the prediction of their specific biological functions. Some SCPs have been reported to function as virulence factors with diverse modes of action[20]. However, deletion of the SCP$^{Ss}$-homolog in *B. cinerea* (SCP$^{Bc}$/Bcplp1) or its homolog *VmE02* in *V. mali* did not alter pathogenicity on various plants[12,21] (Supplementary Fig. 5b, c), suggesting that these proteins either do not contribute significantly to virulence or act redundantly with others during infection. To assess whether SCP$^{Ss}$ homologs from other fungi and oomycetes display RLP30-dependent PAMP activity, homologs from *B. cinerea* (SCP$^{Bc}$) and *Phytophthora infestans* (SCP$^{Pi}$) were produced in *P. pastoris* (Supplementary Fig. 1e). Similar to SCP$^{Ss}$, SCP$^{Bc}$ and SCP$^{Pi}$ induced the production of ethylene and MAPK activation in wild-type or *rlp30-2*/RLP30-YFP plants, but not in *rlp30-2*, *sobir1-12* or *bak1−5 bkk1−1* mutants or SCP$^{Ss}$-insensitive accessions (Fig. 2a and Supplementary Fig. 5d, e). Together, these data demonstrate that SCP$^{Ss}$ homologs, common among fungi and oomycetes, are perceived by the RLP30/SOBIR1/BAK1 receptor complex in *Arabidopsis*.

### SCP$^{Ss}$ is sensed by RE02 in *N. benthamiana*

SCP$^{Ss}$ recognition systems are present in both Brassicaceae and Solanaceae (Fig. 1d), yet, reciprocal sequence similarity searches suggest absence of RLP30 from Solanaceae. The *N. benthamiana* receptor-like protein RE02 (RESPONSE TO VmE02) was recently identified to mediate the perception of VmE02, a cysteine-rich PAMP derived from *V. mali* with 55% identity to SCP$^{Ss}$[12,22]. To compare the ligand specificity of RE02 with RLP30, *RE02* expression in *N.*

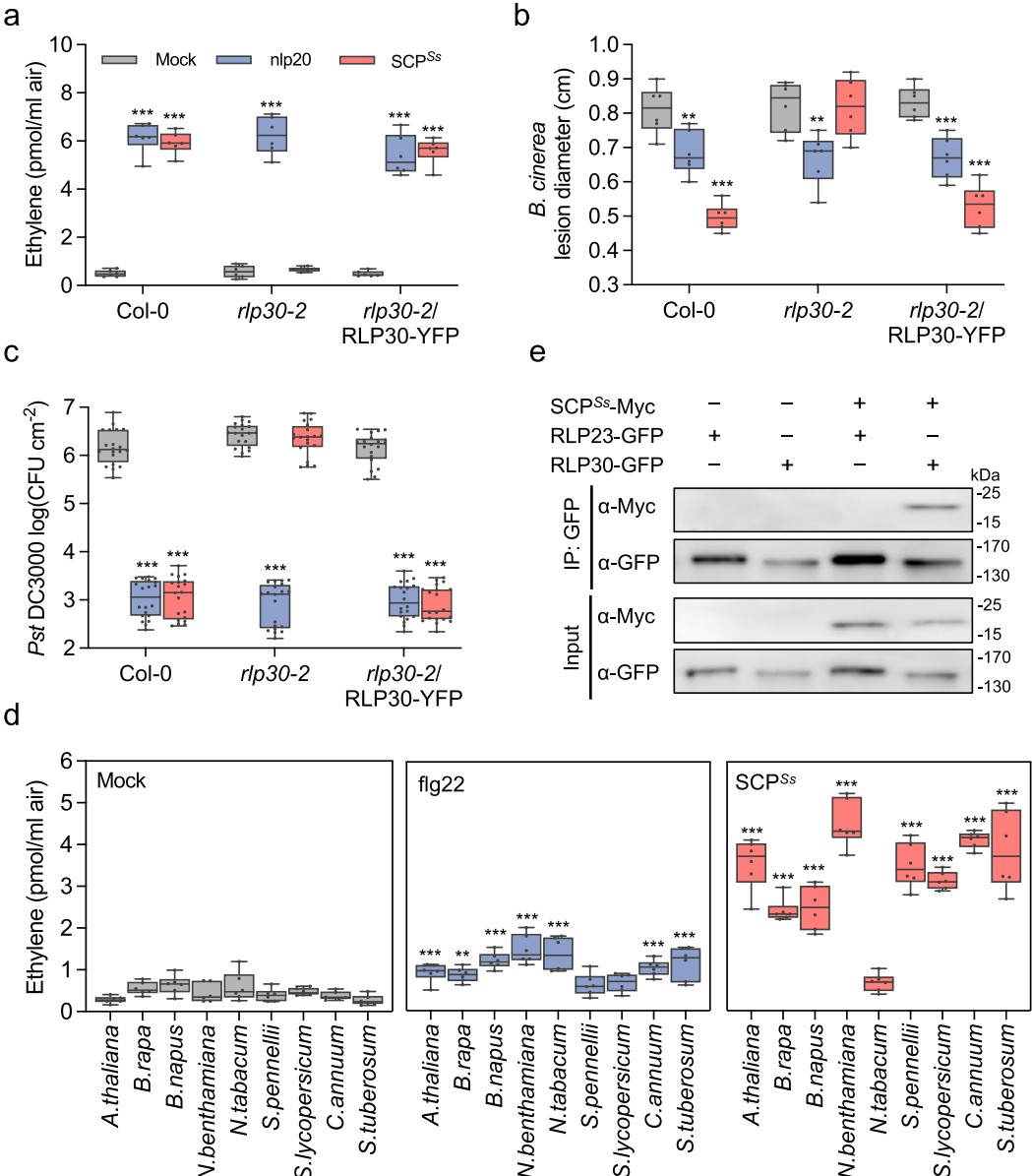

**Fig. 1 | SCP$^{Ss}$ is immunogenic and binds to RLP30. a** Ethylene accumulation in *Arabidopsis* Col-0 wild-type plants, *rlp30-2* mutants, or an *rlp30-2* line complemented with a *p35S::RLP30-YFP* construct 4 h after treatment with water (mock), 1 μM nlp20, or 1 μM *P. pastoris*-expressed SCP$^{Ss}$. **b** *B. cinerea* infected area as determined by lesion diameter on day 2 after 24 h-treatment of Col-0 wild-type plants, *rlp30-2* mutants or the *rlp30-2/RLP30-YFP* complementation line with water (mock), 1 μM nlp20, or 1 μM SCP$^{Ss}$. **c** Bacterial growth in plants pre-treated with water (mock), 1 μM nlp20, or 1 μM SCP$^{Ss}$ 24 h before infiltration of *Pst* DC3000. Bacteria (colony forming units, CFU) were quantified in extracts of leaves 3 days after inoculation. **d** Ethylene accumulation in plants of the Brassicaceae and Solanaceae family 4 h after treatment with water (mock), 1 μM flg22, or 1 μM SCP$^{Ss}$.

**e** Ligand-binding assay in *N. benthamiana* transiently co-expressing SCP$^{Ss}$-myc and either RLP30-GFP or RLP23-GFP. Leaf protein extracts (Input) were used for co-immunoprecipitation with GFP-trap beads (IP:GFP) and immunoblotting with tag-specific antibodies. For **a**–**d**, data points are indicated as dots (n = 6 for **a**, **b**, **d**; n = 20 for **c**) and plotted as box plots (center line, median; bounds of box, the first and third quartiles; whiskers, 1.5 times the interquartile range; error bar, minima and maxima). Statistically significant differences from mock treatments in the respective plants are indicated (two-sided Student's *t* test, **P ≤ 0.01, ***P ≤ 0.001). Source data are provided as a Source data file. All assays were performed at least three times with similar results.

*benthamiana* was targeted by virus-induced gene silencing (VIGS), resulting in almost complete loss of SCP$^{Ss}$-induced ethylene production (Fig. 2b and Supplementary Fig. 6a). In contrast, leaves of silenced plants, either transiently transformed with *RE02*-GFP or *RLP30*-GFP exhibited clear SCP$^{Ss}$ sensitivity (Fig. 2b and Supplementary Fig. 6b). Likewise, stable expression of *RE02*–GFP also reconstituted responsiveness to SCP$^{Ss}$ in *Arabidopsis rlp30-2* mutants (Fig. 2c and Supplementary Fig. 6c). Hence, SCP$^{Ss}$ recognition is mediated by RE02 in *N. benthamiana*.

## RLP30 and RE02 evolved independently for at least 140 million years

RLP30 and RE02 recognize the same microbial pattern but share only about 24% sequence identity (Supplementary Fig. 4b). To investigate RLP30 and RE02 evolution, we analyzed the phylogenetic distribution of receptor proteins related to RLP30. We calculated a detailed phylogeny based on a dataset comprising *A. thaliana* protein sequences, those of other Brassicaceae, Fabaceae, Solanaceae, grape vine (*Vitis vinifera*, member of the sister lineage to rosids and asterids), and

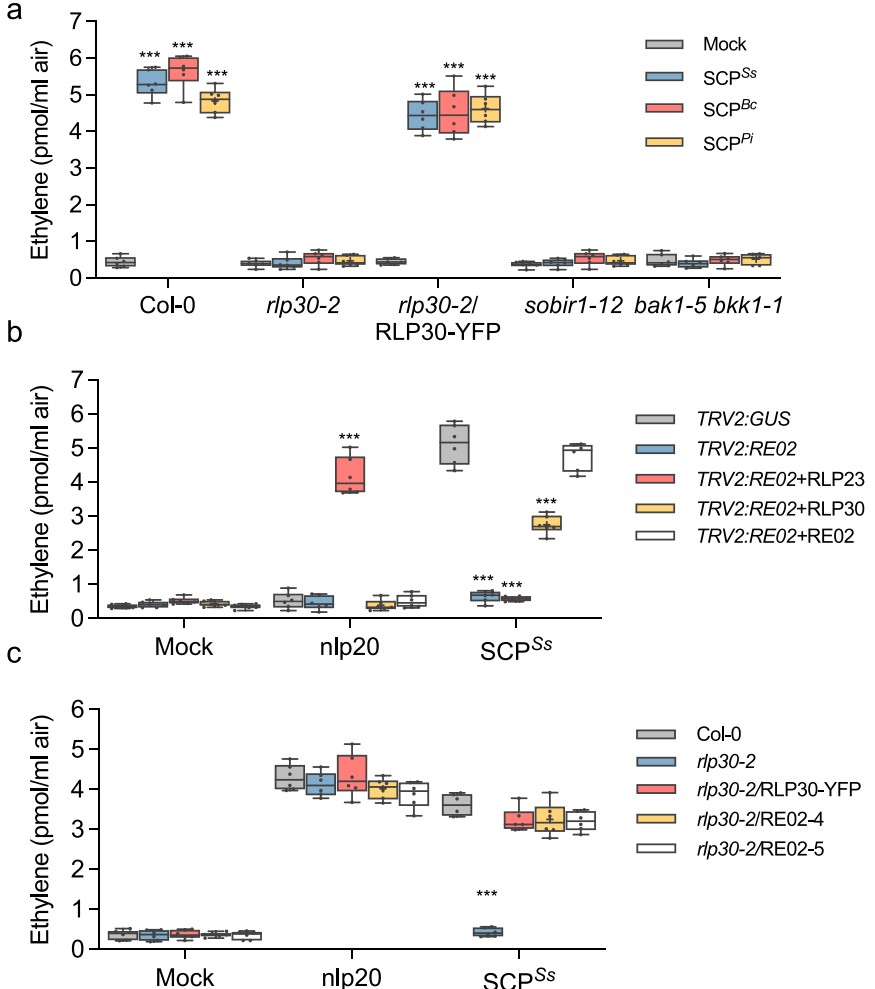

**Fig. 2 | SCPs are sensed by RE02 and RLP30. a** Ethylene accumulation in *Arabidopsis* wild-type plants (Col-0) or indicated mutants 4 h after treatment with water (mock), 1 μM nlp20, 1 μM SCP$^{Ss}$, or homologs from *B. cinerea* (*Bc*) and *Phytophthora infestans* (*Pi*). **b** *N. benthamiana* plants were silenced for *RE02*, and transiently transformed with *RLP23-GFP*, *RLP30-GFP*, or *RE02-GFP*, respectively. The *TRV2:GUS* construct was used as a control. Ethylene production was measured 4 h after treatment with water (mock), 1 μM nlp20, or 1 μM SCP$^{Ss}$. **c** Ethylene accumulation in *Arabidopsis rlp30-2* mutants or lines stably expressing RLP30-YFP or RE02-GFP (line 4 and 5) 4 h after treatment with water (mock), 1 μM nlp20, or 1 μM SCP$^{Ss}$. Data points are indicated as dots (*n* = 6) and plotted as box plots (center line, median; bounds of box, the first and third quartiles; whiskers, 1.5 times the interquartile range; error bar, minima and maxima). Statistically significant differences from control treatments are indicated (two-sided Student's *t* test, ***$P \le 0.001$). Source data are provided as a Source data file. Each experiment was repeated three times with similar results.

*Amborella trichopoda* (the sister species to all other angiosperms) (Supplementary Fig. 7). Our data strongly support a sister group relationship of the RE02-like subfamily and the RLP30 subfamily with both subfamilies including members from *Amborella*. This indicates a split of the RLP30 and RE02 subfamilies in the lineage leading to all flowering plants, allowing these subfamilies to evolve independently for least 140 million years[23]. Hence, SCP sensor systems arose independently from convergent evolution.

Interestingly, the RE02-like subfamily does not include any malvid protein sequences, suggesting that malvids lost *RE02*-like genes, while in contrast RE02-like genes were copied multiple times in grape vine (Supplementary Fig. 7). Thus, we speculate that, compared to Solanaceae, immunity against microbial pathogens in malvids, including Brassicaceae or Fabaceae, might rely more strongly on *RLP30*-related genes.

Unlike the RE02-like subfamily, members of the RLP30-like subfamily were retained in all species analyzed and show lineage-specific expansions, as predicted previously by pre-computed phylogenetic trees from PhyloGene[24]. Within the Brassicaceae, 30 *RLP30*-like genes are encoded by the *A. thaliana* genome. They form a monophyletic

clade with other Brassicaceae genes supported by maximum bootstrap support, suggesting that they originate from a single gene copy in the lineage leading to Brassicaceae.

## Distinct perception of SCPs in Brassicaceae and Solanaceae

Immunogenic activity of SCP$^{Ss}$ is sensitive to treatment with DTT (Supplementary Fig. 8a), suggesting that perception by the RLPs RLP30 and RE02 depends on the tertiary structure of SCP$^{Ss}$. The SCP$^{Ss}$ protein is predicted to form four disulfide bonds, most likely connecting cysteine residues C1 and C4, C2 and C3, C5 and C8 and C6 and C7, respectively (Fig. 3a). Replacing individual cysteine residues in SCP$^{Ss}$ with serine leads to loss of activity and RLP30 complex formation in *Arabidopsis* for all eight substitutions (Fig. 3b and Supplementary Fig. 8b, c). When testing the individual Cys substitutions for ethylene-inducing activity in *N. benthamiana* or tomato, we observed that only the four N-terminal Cys residues, forming the bonds C1-C4 and C2-C3, were required for ethylene-induction, while replacement of the other four Cys residues did not affect the activity (Fig. 3b and Supplementary Fig. 8d). Hence, whereas RLP30 only senses intact SCP$^{Ss}$, RE02-mediated perception requires only part of the SCP$^{Ss}$ tertiary structure.

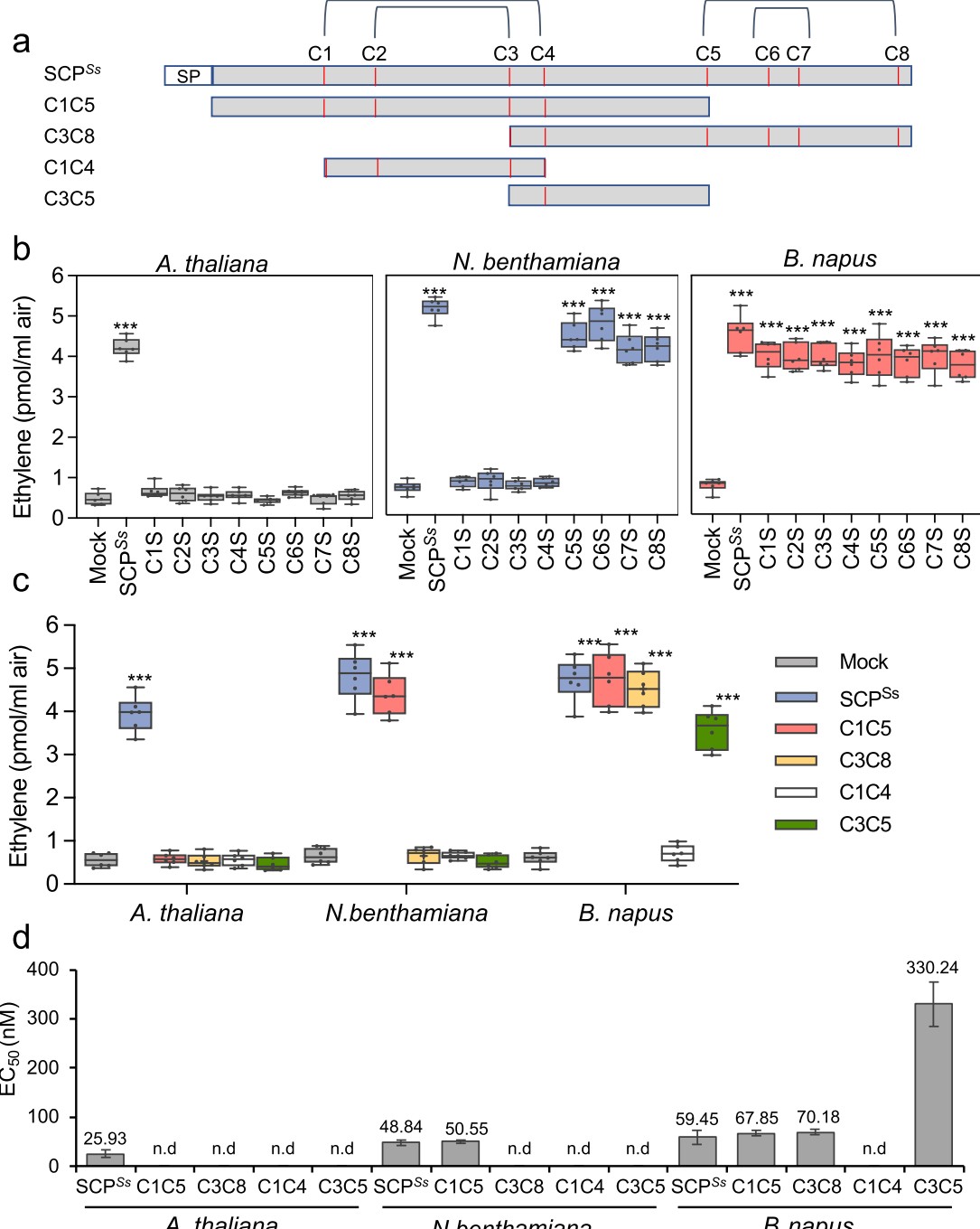

**Fig. 3 | SCP$^{Ss}$ sensing differs in Brassicaceae and Solanaceae. a** Schematic representation of SCP$^{Ss}$ and derived truncations C1C5, C3C8, C1C4 and C3C5. Cysteine (C) residues are numbered and indicated as red lines, predicted disulfide bridges (DB, http://clavius.bc.edu/~clotelab/DiANNA/) are shown on top. SP, signal peptide. **b, c** Ethylene accumulation in *A. thaliana* Col-0, *N. benthamiana* or *B. napus* wild-type plants after 4 h treatment with water (mock), 1 μM SCP$^{Ss}$, and SCP$^{Ss}$ with individual cysteine to serine mutations (C1S to C8S, according to cysteine numbering shown in **a**) (**b**) or SCP$^{Ss}$ truncations depicted in **a** (**c**). Data points are indicated as dots (*n* = 6) and plotted as box plots (center line, median; bounds of box, the first and third quartiles; whiskers, 1.5 times the interquartile range; error bar, minima and maxima). Statistically significant differences from mock treatments in the respective plants are indicated (two-sided Student's *t* test, ***$P ≤ 0.001$). **d** Determination of EC$_{50}$ values of SCP$^{Ss}$ and the various SCP$^{Ss}$ truncations using ethylene accumulation in *A. thaliana*, *N. benthamiana*, or *B. napus* plants after treatment with increasing concentrations of recombinant SCP$^{Ss}$ versions (produced in *Pichia*) or synthetic C3C5 (n.d, activity not detectable). Source data are provided as a Source data file. Each experiment was repeated three times with similar results.

Consistent with the results of the individual cysteine replacements, a truncated form of SCP$^{Ss}$, comprising C1 to C5 (C1C5), was active in tomato and *N. benthamiana* but not in *Arabidopsis* (Fig. 3c, d and Supplementary Fig. 8e). C1C4, a variant of the protein further truncated from both the N- and the C-terminus but still comprising the first four Cys residues, proved inactive in *Arabidopsis* but also in *N.*

*benthamiana* and tomato (Fig. 3c, d and Supplementary Fig. 8e), indicating that in addition to the disulfide bonds also amino acids flanking the C1C4 peptide are important for immunogenic activity in solanaceous plants. Thus, SCP$^{Ss}$ is one of the very few identified ligands whose tertiary structure is important for PAMP activity similar to IF1-recognition by RLP32[14].

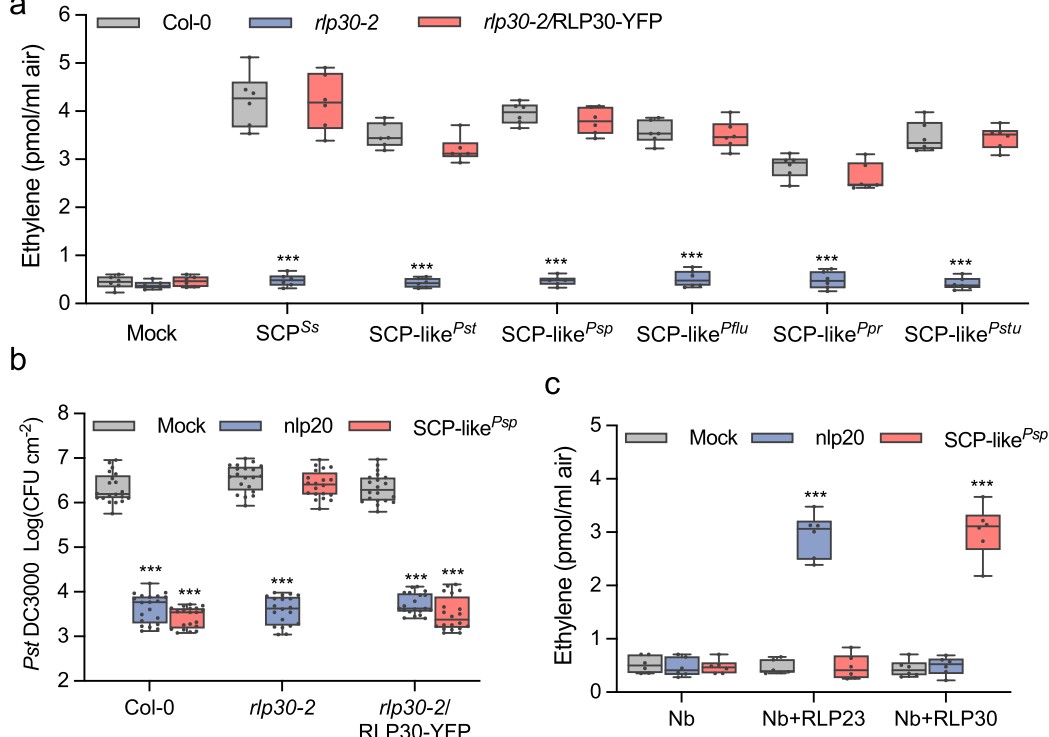

**Fig. 4 | Pseudomonads produce a RLP30-dependent elicitor activity. a** Ethylene accumulation in Col-0 wild-type plants, *rlp30-2* mutants or an *rlp30-2* line complemented with a *p35S::RLP30-YFP* construct 4 h after treatment with water (mock), 1 µM SCP$^{Ss}$ or 1.5 µg/ml SCP-like from *Pseudomonas syringae* pv. *tomato* (*Pst*), *P. syringae* pv. *phaseolicola* (*Psp*), *P. fluorescens* (*Pflu*), *P. protegens* (*Ppr*), and *P. stutzeri* (*Pstu*). **b** Bacterial growth in Col-0 wild-type plants, *rlp30-2* mutants, or the *rlp30-2/RLP30-YFP* complementation line treated with water (mock), 1 µM nlp20, or 1.5 µg/ml SCP-like$^{Psp}$ 24 h before infiltration of *Pst* DC3000. Bacteria (colony forming units, CFU) were quantified in extracts of leaves 3 days after inoculation. **c** Ethylene

accumulation in *N. benthamiana* (*Nb*) plants transiently expressing RLP23-GFP or RLP30-GFP and treated for 4 h with water (mock), 1 µM nlp20, or 1.5 µg/ml SCP-like$^{Psp}$. Data points are indicated as dots (*n* = 6 for **a**, c; *n* = 20 for **b**) and plotted as box plots (center line, median; bounds of box, the first and third quartiles; whiskers, 1.5 times the interquartile range; error bar, minima and maxima). Statistically significant differences from Col-0 plants (**a**) or mock treatments (**b**, **c**) are indicated (two-sided Student's *t* test, ***$P \leq 0.001$). Source data are provided as a Source data file. Each experiment was repeated three times with similar results.

Intriguingly, all SCP$^{Ss}$ mutants exhibited elicitor activity comparable to the wild-type SCP$^{Ss}$ in *B. napus*, *B. rapa*, and *B. oleracea* (Fig. 3b–d and Supplementary Fig. 8d, e), indicating that in these Brassicaceae species SCP$^{Ss}$-immunogenic activity is independent of cysteine-bridges and, thus, of its tertiary structure. Hence, at least two distinct SCP$^{Ss}$-perception systems are present in Brassicaceae, one responding to intact SCP$^{Ss}$ protein (as in *Arabidopsis*), and the other one responding to a SCP$^{Ss}$-derived peptide (as in *Brassica* ssp.). As both C1C5 and C3C8 displayed activity in Brassicaceae, we generated the 36-amino-acid peptide C3C5 spanning the overlapping region of C1C5 and C3C8 (Fig. 3a). C3C5, which contains only one cysteine residue (C4) and thus cannot form any disulfide bridge, was found to be nearly as active as intact SCP$^{Ss}$ in *B. napus*, *B. rapa*, and *B. oleracea* (Fig. 3c, d and Supplementary Fig. 8e), indicating that this SCP$^{Ss}$-fragment is sufficient to trigger immune responses in *Brassica* species. We therefore demonstrate that unlike RLP30-mediated recognition of intact SCP$^{Ss}$ in *Arabidopsis*, *Brassica* species can sense a small SCP$^{Ss}$ epitope that is distinct from the disulfide-bond containing peptide perceived in Solanaceae. In general, such large differences of sensor systems rather occur in distantly related plant families. However, similar to SCP$^{Ss}$, PG perception also differs within the Brassicaceae family and *A. thaliana*, *A. arenosa*, and *B. rapa* perceive immunogenic PG fragments pg9(At), pg20(Aa) and pg36(Bra), respectively[15]. Thus, the perception diversity for SCP$^{Ss}$ or PG in closely related species within the same plant family could indicate adaptive processes in plant pathogen co-evolution in Brassicaceae.

## RLP30 senses *Pseudomonas*-derived PAMPs

Intriguingly, *rlp30* mutants have been found to be more susceptible to bacterial infection with *P. syringae* pv. *phaseolicola* (*Psp* 1448A)[25]. We thus wondered whether RLP30 might also recognize a PAMP from Pseudomonads, despite the apparent absence of SCP$^{Ss}$ homologs in bacteria[12]. Using the chromatographic enrichment protocol established for SCFE1[18], we could detect elicitor-activity in culture media of different *Pseudomonas* species, such as *Psp* 1448A, *Pst* DC3000, *P. fluorescens*, *P. protegens*, and *P. stutzeri* (Fig. 4a). Focusing on the activity from *Psp*, termed SCP-like$^{Psp}$, we observed typical immune responses such as ethylene production, MAPK activation, as well as priming for resistance towards *Pst* DC3000 infection in *Arabidopsis* Col-0 wild-type plants and the *rlp30-2* mutant complemented with RLP30 (Fig. 4b and Supplementary Fig. 9a). SCP-like$^{Psp}$ also induced ethylene biosynthesis in *fls2 efr* and *rlp32-2* mutants, indicating that activity was not due to the well-established bacterial PAMPs flagellin, EF-Tu or IF1 (Supplementary Fig. 9b). In contrast, SCP-like$^{Psp}$ had no activity in the mutant *rlp30-2*, in nine of the SCP$^{Ss}$-insensitive accessions, and in the *sobir1-12* and *bak1-5 bkk1-1* mutants (Fig. 4a and Supplementary Fig. 9c, d), suggesting that perception of SCP-like$^{Psp}$ involves all the components required for perception of SCP$^{Ss}$. So far, our attempts to determine the molecular identity of SCP-like$^{Psp}$ were not successful. However, much like SCP$^{Ss}$, the immunogenic activity of SCP-like$^{Psp}$ was sensitive to treatment with DTT and proteases, but resistant to heat and SDS treatment, suggesting that it is likely also a protein stabilized by disulfide bridges (Supplementary Fig. 9e).

## RLP30 and RE02 differ in perception of SCP-like$^{Psp}$

SCP-like$^{Psp}$ showed activity in *A. thaliana*, *B. rapa*, and *B. oleracea*. Thus, species of the *Brassicaceae* family perceive SCPs and therefore have to encode functional RLP30 homologs (Supplementary Fig. 9f). Interestingly, none of the species from the Solanaceae family tested showed responsiveness to bacterial SCP-like$^{Psp}$ (Supplementary Fig. 9f). These species included *N. benthamiana*, *S. pennellii*, *S. lycopersicum*, *S. tuberosum* and *C. annuum* with functional SCP$^{Ss}$ perception, indicating that RE02 and solanaceous homologs do not respond to SCP-like$^{Psp}$. Ectopic expression of RLP30 in *N. benthamiana* conferred sensitivity to SCP-like$^{Psp}$, but not to nlp20 which required expression of RLP23 (Fig. 4c and Supplementary Fig. 9g). Thus, RLP30, but not RE02, senses patterns from three microbial kingdoms.

## RLP30 expression in *N. tabacum* confers increased resistance to bacterial, fungal and oomycete pathogens

Since RLP30 has the property of sensing PAMPs from microorganisms of three kingdoms, we introduced RLP30 into SCP$^{Ss}$-insensitive *N. tabacum* plants. We observed that *N. tabacum* plants transiently expressing RLP30-GFP were responsive to SCP$^{Ss}$ and SCP-like$^{Psp}$, albeit with only a moderate increase in production of ethylene (Supplementary Fig. 10a, b). We wondered whether this might be due to a partial incompatibility of RLP30 with the endogenous tobacco SOBIR1. Indeed, co-expression of RLP30 with *At*SOBIR1 in *N. tabacum* established higher responses to SCP$^{Ss}$ and SCP-like$^{Psp}$, whereas plants co-transformed with *RLP30* and *NbSOBIR1* or *SlSOBIR1* showed responses as plants without additional SOBIR1 (Supplementary Fig. 10a, b). Thus, we introduced *RLP30-RFP* together with the *AtSOBIR1-GFP* into *N. tabacum* and selected two stable transgenic lines (*#49* and *#55*) based on detectable expression of RLP30 and *At*SOBIR1 (Supplementary Fig. 10c). Neither of the two transgenic lines exhibited autoimmune phenotypes as mock treatment did not induce ethylene accumulation and no elevated expression of salicylic acid marker gene *PR-1a* or jasmonic acid marker gene *PDF1.2* could be detected in untreated plants (Fig. 5a and Supplementary Fig. 10a, d). In contrast to wild-type tobacco plants, *#49* and *#55* transgenic lines clearly responded to SCPs or SCP-like$^{Psp}$ (Fig. 5a). Importantly, when inoculated with the host-adapted, virulent bacterial pathogen *P. syringae* pv. *tabaci*, both transgenic lines exhibited less bacterial growth than wild-type plants (Fig. 5b). Moreover, compared to wild-type plants, both transgenic lines also exhibited less symptom development upon infection with the necrotrophic fungus *B. cinerea* and the oomycete *Phytophthora capsici* (Fig. 5c, d). Taken together, stable, ectopic expression of RLP30 in *N. tabacum* conferred sensitivity to SCPs and SCP-like$^{Psp}$ and rendered these transgenic plants more resistant to destructive bacterial, fungal, and oomycete pathogens.

## Discussion

Plants harbor a tremendous number of potential receptor genes, however, their characterization is strictly dependent on knowledge of respective ligands. We identified here the ligand of the previously described RLP30[18] as a secreted, small cysteine-rich protein (SCP). SCP$^{Ss}$ is widely distributed across fungi and oomycetes[12] and its tertiary structure is important for PAMP activity, features that are rare and as combination unprecedented among known PAMPs. Additionally, SCP perception also shows unique characteristics. Supplementary Fig. 11 summarizes our findings showing that SCP$^{Ss}$ and related elicitors do not only serve as ligands for RLP30 in *Arabidopsis*, but also for RE02 in *N. benthamiana* and a yet unknown *Brassica* PRR. RLP30 and RE02 share low similarity and all these PRRs moreover show distinct requirements for SCP$^{Ss}$: RLP30 requires intact SCP$^{Ss}$, Solanaceae receptors including RE02 sense the N-terminal part containing only two disulfide bonds and for SCP$^{Ss}$ perception in *Brassica* (except *Arabidopsis*) a fragment of 36 amino acid is sufficient. Hence, the distinct perception mechanisms of the same fungal protein most likely arose

through convergent evolution of LRR-RPs in different plant families. Convergent evolution of RLP30 and RE02 is supported by our phylogenetic analysis, indicating that RLP30 and RE02 subfamilies evolved independently for at least 140 million years. Moreover, RE02 and RLP30 have different protein structures with RE02 harboring 28 LRR motifs versus 21 LRRs in RLP30 and both proteins have highly divergent island domains. Based on the motif features within the island domain, RE02 (Kx$_5$Y motif) and RLP30 (Hx$_8$KG motif) are classified into two different clades[26]. Notably, the island domain of plant LRR-proteins was shown to participate in ligand binding[15,27], explaining the different structural requirements of RLP30 and RE02 for SCP$^{Ss}$ perception and the ability of RLP30 to recognize a bacterial SCP-like pattern.

Intriguingly, despite the broad capacity of multiple Brassicaceae and Solanaceae species to detect SCP$^{Ss}$, multiple *Arabidopsis* ecotypes carry altered RLP30 receptors that do not recognize SCP$^{Ss}$. Although RLP30 is considered a conserved LRR-RP and RLP30 sequences are present in most *Arabidopsis* ecotypes[28], slight sequence variations might lead to functional diversification. Alternatively, accession-specific loss of RLP30 function might be explained by redundancy of *Arabidopsis* cell-surface immune receptors for fungal patterns such as SCPs, NLPs, PGs, oligogalacturonides or chitin perceived by *Arabidopsis* receptors RLP30, RLP23, RLP42, WAK1 or LYK5/CERK1, respectively[19,29–31]. This complexity is also reminiscent of *Arabidopsis* sensors for *Pseudomonas*-derived patterns[4].

SCP$^{Ss}$-like sequences are not found in bacteria[12], yet RLP30, but not RE02, confers recognition of SCP-like$^{Psp}$ from both pathogenic and beneficial *Pseudomonas* species. Hence, RLP30 is a unique receptor that can recognize most likely distinct ligands from fungi/oomycetes and bacteria. Such a broad detection range predestines PRRs such as RLP30 as a valuable genetic tool to boost crop immunity, and RLP30 expression in *N. tabacum* does indeed confer quantitative resistance to bacterial, fungal, and oomycete infections. Notably, RE02 also functions in *Arabidopsis*, indicating that the complex partner proteins are sufficiently conserved across wide taxonomic space, which is also supported by multiple reports on successful transfer of PRRs across taxa[14,19,32,33]. As RLP30, however, only modestly functions in tobacco without co-expressing AtSOBIR1, we propose that co-expression of heterologous PRR/co-receptor pairs may be deployed to overcome putative incompatibilities of ectopically expressed PRRs. Consequently, this way enhanced broad-spectrum resistance may be engineered as compared to expressing single PRRs alone.

## Methods

### Plant materials and growth conditions

*Arabidopsis thaliana* natural accessions[34,35] and mutant plants were grown in soil for 7-8 weeks in a growth chamber under short-day conditions (8 h photoperiod, 22 °C, 40–60% humidity). Mutant lines and derived complemented transgenic lines are listed in Supplementary Table 2. Solanaceae and *Brassica* plants were grown in a greenhouse at 23 °C under long-day conditions of 16 h of light and 60–70% humidity.

### Peptides

Synthetic peptides flg22, nlp20, and SCP$^{Ss}$C3C5 (SAYTCAA-PAKSHLTAESDYWVFYWGNEGVSPGVGST) (GenScript) were prepared as 10 mM stock solutions in 100% dimethyl sulfoxide (DMSO) and diluted in water to the desired concentration before use.

### SCP$^{Ss}$ identification

Partially purified SCFE1 fractions from *S. sclerotiorum* strain 1946 were used[18]. The most active fractions from eight rounds of fungal culture and the two-step cation-exchange chromatography purification protocol were pooled and dialyzed overnight in 2 l 25 mM MES, pH 7.0, at 4 °C in a dialysis membrane (ZelluTrans,

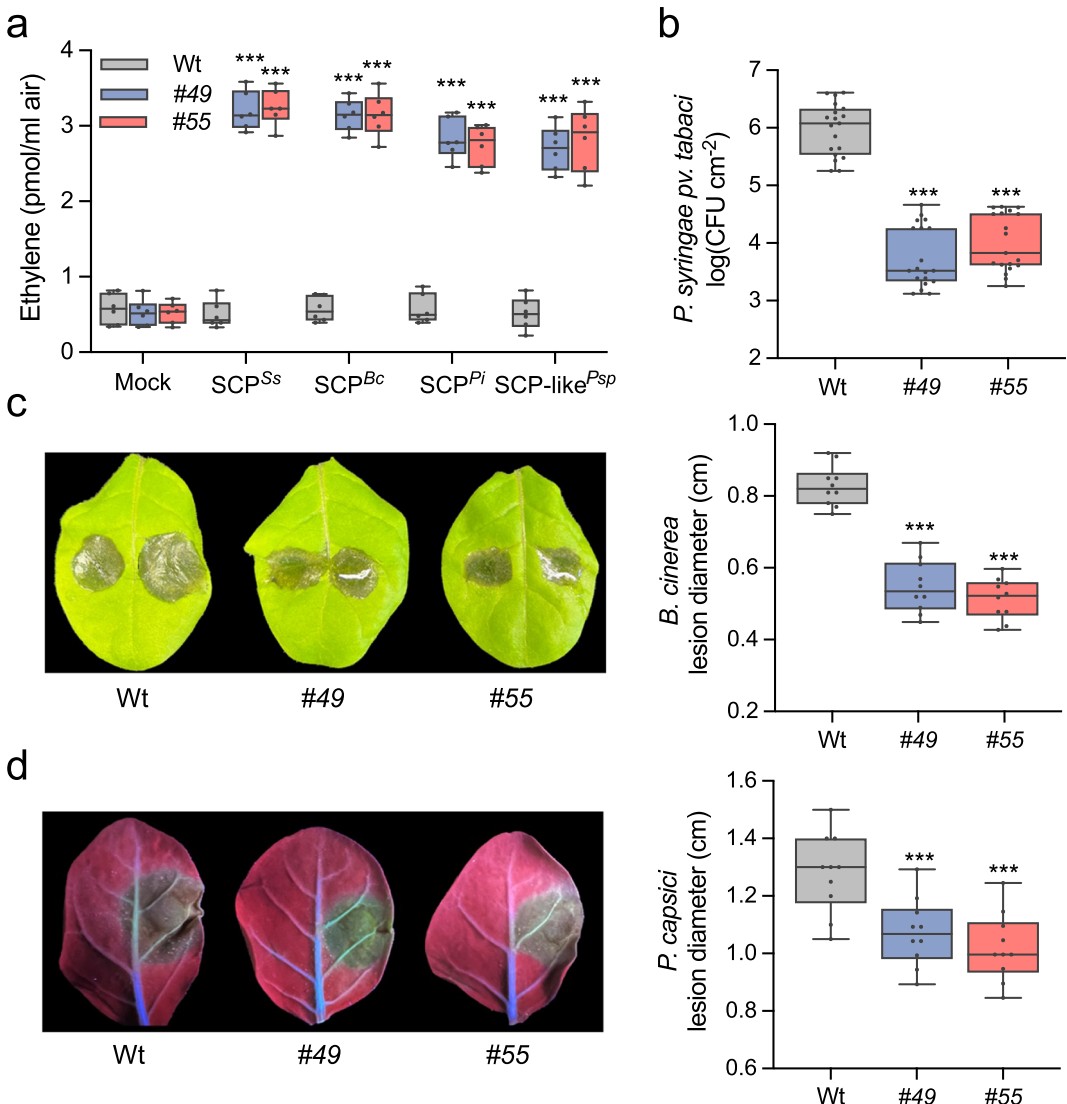

**Fig. 5 | RLP30 expression in *N. tabacum* confers increased resistance to bacterial, fungal, and oomycete pathogens. a** Ethylene accumulation in *N. tabacum* wild-type plants (Wt) or two transgenic lines (*#49* and *#55*) stably expressing RLP30-RFP and AtSOBIR1-GFP after 4 h treatment with water (mock), 1 μM SCP from indicated sources, or 1.5 μg/ml SCP-like^Psp. **b** Bacterial growth of *P. syringae* pv. *tabaci* in *N. tabacum* wild-type plants (Wt) or transgenic *RLP30/SOBIR1* lines (*#49* and *#55*). Bacteria (colony forming units, CFU) were quantified in extracts of leaves 3 days after inoculation. **c** *B. cinerea* infected area on leaves of *N. tabacum* wild-type plants (Wt) or transgenic *RLP30/SOBIR1* lines (left, shown are representative leaves) and determination of lesion diameter on day 2 after drop inoculation (right).

**d** Growth of *Phytophthora capsici* on leaves of *N. tabacum* wild-type plants (Wt) or transgenic *RLP30/SOBIR1* lines by determination of lesion size (right) of lesions observed under UV light (left, shown are representative leaves) on day 2 after drop inoculation. Data points are indicated as dots (*n* = 6 for **a**; *n* = 20 for **b**, *n* = 10 for **c**, **d**) and plotted as box plots (center line, median; bounds of box, the first and third quartiles; whiskers, 1.5 times the interquartile range; error bar, minima and maxima). Statistically significant differences from wild-type (Wt) plants are indicated (two-sided Student's *t* test, ***$P \leq 0.001$). Source data are provided as a Source data file. Each experiment was repeated three times with similar results.

nominal MWCO: 3.5; 46 mm, Roth), afterwards vacuum-filtrated through a cellulose acetate membrane (Ciro Manufacturing Corporation, pore size: 0.2μm) and loaded with a flow rate of 1 ml/min onto a Source 15S 4.6/100PE column (GE Healthcare) equilibrated with buffer A (50 mM MES, pH 5.4). After washing with buffer A, bound proteins were eluted with a linear gradient of buffer B (500 mM KCl, 50 mM MES, pH 5.4; 0–50% in 15 column volumes) and 500 μl fractions were collected using automated fractionation.

**LC-MS/MS sample preparation**

Three elicitor-active (B11 to B13, Supplementary Fig. 1b), and two inactive (B9 and B15) fractions of SCFE1 (in 50 mM MES buffer pH 7.0 with varying concentration of KCl, depending on the elution time point) were selected for proteomics analysis. For each

fraction a sample volume of 88 μl was used, which corresponded to a total protein content per fraction from 3.9 μg (B9) to 16.0 μg (B13), as determined using the BCA protein assay kit (Thermo Fisher Scientific). All samples were reduced (10 mM DTT, 30 min, 30 °C) and carbamidomethylated (55 mM CAA, 30 min, room temperature). Digestion of proteins was carried out by addition of trypsin (proteomics grade, Roche) at a 1/50 enzyme/protein ratio (w/w) and incubation at 37 °C overnight. Digests were acidified by addition of 0.5% (v/v) formic acid, and desalted using self-packed StageTips (three disks per micro-column, ø 1.5 mm, C18 material, 3 M Empore). The peptide eluates were dried to completeness and stored at −80 °C. For LC-MS/MS analysis all samples were resuspended in 11 μl 0.1% formic acid in HPLC grade water and 5 μl sample per measurement was injected into the mass spectrometer.

## LC-MS/MS data acquisition

LC-MS/MS measurements were performed on a nanoLC Ultra1D+ (Eksigent, Dublin, CA) coupled online to a Q-Exactive HF-X mass spectrometer (Thermo Fisher Scientific). Peptides were loaded onto a trap column (ReproSil-pur C18-AQ, 5 μm, Dr. Maisch, 20 mm × 75 μm, self-packed) at a flow rate of 5 μl/min in 100% solvent A (0.1% formic acid in HPLC grade water). Subsequently, peptides were transferred to an analytical column (ReproSil Gold C18-AQ, 3 μm, Dr. Maisch, 400 mm × 75 μm, self-packed) and separated using a 50 min linear gradient from 4% to 32% of solvent B (0.1% formic acid in acetonitrile and 5% (v/v) DMSO) at 300 nl/min flow rate. Both nanoLC solvents contained 5% (v/v) DMSO to boost the nanoESI response of peptides. The eluate from the analytical column was sprayed via a stainless-steel emitter (Thermo Fisher Scientific) at a source voltage of 2.2 kV into the mass spectrometer. The transfer capillary was heated to 275 °C. The Q-Exactive HF-X was operated in data-dependent acquisition (DDA) mode, automatically switching between MS1 and MS2 spectrum acquisition. MS1 spectra were acquired over a mass-to-charge ($m/z$) range of $m/z$ 350–1400 at a resolution of 60,000 using a maximum injection time of 45 ms and an AGC target value of 3e6. Up to 18 peptide precursors were isolated (isolation window $m/z$ 1.3, maximum injection time 25 ms, AGC value 1e5), fragmented by high-energy collision-induced dissociation (HCD) using 26% normalized collision energy and analyzed at a resolution of 15,000 with a scan range from $m/z$ 200–2000. Precursor ions that were singly charged, unassigned, or with charge states >6+ were excluded. The dynamic exclusion duration of fragmented precursor ions was 25 s.

## LC-MS/MS data analysis

Peptide identification and quantification was performed using Max-Quant (version 1.5.3.30)[36]. MS2 spectra were searched against a *Sclerotinia sclerotiorum* protein database (assembly accession number PRJNA348385, 11130 protein sequence entries), supplemented with common contaminants (built-in option in MaxQuant). Carbamidomethylated cysteine was set as fixed modification and oxidation of methionine and N-terminal protein acetylation as variable modifications. Trypsin/P was specified as proteolytic enzyme. Precursor tolerance was set to 4.5 ppm, and fragment ion tolerance to 20 ppm. Results were adjusted to 1% false discovery rate (FDR) on peptide spectrum match (PSM) and protein level employing a target-decoy approach using reversed protein sequences. The minimal peptide length was defined as 7 amino acids, the "match-between-run" function was enabled (default settings). To assess and quantitatively compare the concentrations of the detected proteins the "intensity based absolute quantification" (iBAQ)[37] algorithm was employed. The detected 22 proteins were further filtered and prioritized according to the following characteristics: (1) good correlation between the elicitor activity profile and the measured mass spectrometric protein intensity over the five investigated SCFE1 fractions; (2) presence of a predicted signal peptide sequence (using the SignalP v.6.0 server)[38]; (3) protein molecular weight in the range between 10 to 30 kDa; (4) high cysteine content relative to protein length; (5) detection of at least two unique peptides per protein (Supplementary Table 1).

## SCP$^{Ss}$ expression and purification

For expression of SCP$^{Ss}$ in *N. benthamiana* leaves, the SCP$^{Ss}$ sequence was cloned into the pBIN-plus vector with a C-terminal GFP tag[15]. *Agrobacterium tumefaciens* strain GV3101 carrying pBIN-plus-SCP$^{Ss}$ was infiltrated into leaves of 4–6-week-old *N. benthamiana* plants. At 48 h post agro-infiltration, leaves were vacuum-infiltrated with 0.5 M NaCl, 0.4 M Na$_2$HPO$_4$ and 0.4 M NaH$_2$PO$_4$ and the apoplastic wash-fluid containing SCP$^{Ss}$ (SCP$^{Ss}$-AWF) was collected by centrifugation at 1500 × $g$ for 20 min and 4 °C.

For recombinant protein expression in *Pichia pastoris* KM71H (Multi-Copy Pichia Expression Kit Instructions, Thermo Fisher

Scientific), constructs encoding SCP$^{Ss}$, SCP$^{Bc}$, SCP$^{Pi}$, truncated SCP$^{Ss}$ versions and cysteine-to-serine replacements (all without signal peptide and Stop codon) were cloned into the secretory expression plasmid pPICZalphaA (Thermo Fisher Scientific). Individual cysteine to serine replacements in SCP$^{Ss}$ were amplified from synthetic genes (gBlocks, Integrated DNA Technologies) using primers extended by restriction sites *Eco*RI and *Bam*HI and cloned into pPICZalphaA. Sequence information on primers and synthetic genes used in this study can be found in Supplementary Tables 3 and 5. Protein purification from *P. pastoris* culture medium was achieved by affinity chromatography on 5 ml HisTrapFF column (GE Healthcare; equilibrated in 20 mM Tris-HCl, 200 mM NaCl, pH 8.0), following washing (20 mM Tris-HCl, 200 mM NaCl, 20 mM imidazole, pH 8.0) and elution (buffer gradient 0–500 mM imidazole in equilibration buffer). Recombinant protein expression was verified by concentration determination using the Bradford method and by Western Blot analysis using the anti-His antibody (dilution 1:1000, Abcam).

## SCP-like purification from *Pseudomonas* ssp

*Pseudomonas* strains (*Pseudomonas syringae* pv. *tomato* DC3000, *Pseudomonas syringae* pv. *phaseolicola* 1448A, *Pseudomonas protegens* Pf-5, *Pseudomonas fluorescens* SBW25, *Pseudomonas stutzeri* DSM10701) were grown in baffled flasks (200 ml medium in 1 l flask) in liquid PDB medium (24 g/l Potato Dextrose Broth, Duchefa) for 12 h at 21 °C with 210 rpm shaking. The culture medium was centrifuged at 6000 rpm for 20 min and then vacuum-filtrated through a filter paper (MN 615, Macherey-Nagel) and subjected to a two-step cation exchange chromatography protocol using an ÄKTA Explorer FPLC system (GE Healthcare) kept at 4 °C. In a first step, the culture filtrate was loaded onto HiTrap SP FF column(s) (GE Healthcare) equilibrated in buffer A (50 mM MES, pH 5.4) at a flow rate of up to 15 ml/min. After washing with buffer A, a 100% elution step with buffer B (500 mM KCl, 50 mM MES, pH 5.4) was performed at a flow rate of 1–2 ml/min and the elution peak was monitored with OD280nm, OD254nm and OD214nm and collected manually. The collected eluate was dialyzed against 2 l 25 mM MES, pH 5.4, overnight at 4 °C in a dialysis membrane (Zellu-Trans, nominal MWCO: 3.5; 46 mm, Roth), afterwards vacuum-filtrated through a cellulose acetate membrane (Ciro Manufacturing Corporation, pore size: 0.2 μm) and loaded with a flow rate of 1 ml/min onto a Source 15 S 4.6/100PE column (GE Healthcare) equilibrated with buffer A. After washing with buffer A the bound proteins were eluted with a linear gradient of buffer B (0–50% in 10 column volumes) and 500 μl fractions were collected using automated fractionation.

## Generation of transgenic plants

For the generation of transgenic *rlp30-2* lines expressing *p35S::REO2-GFP*, the coding sequence of *REO2* extended by *Bsa*I cleavage sites (Supplementary Table 3) was cloned into the pGEM®-T vector (Promega). The cauliflower mosaic virus 35 S promoter, *REO2* coding sequence and GFP-tag were assembled via golden gate cloning in the vector LII_F_1-2_-_BB10c[39] and constructs were transformed into *rlp30-2* by floral dipping.

For 35S-promoter-driven stable expression of *RLP30-YFP* in *rlp30-2* mutants or *RLP30-RFP* and *SOBIR1-GFP* in *N. tabacum* L. var. Samsun NN, *RLP30* and *SOBIR1* coding sequences were first cloned into pCR™8/GW/TOPO™ (Thermo Fisher Scientific) and subsequently fused via Gateway cloning to a C-terminal YFP (pB7YWG2[40]), RFP (pB7RWG2[40]) or GFP (pK7FWG2[40]), respectively. Primers used are listed in Supplementary Table 3. For stable transformation, *A. tumefaciens* GV3101 carrying the desired constructs were grown in LB medium with appropriate antibiotics. For *Arabidopsis* transformation, bacterial cells were harvested and resuspended in 5% (w/v) Sucrose, 0.02% (v/v) Silwet at an O.D. of 0.8 and buds of 6–8-week-old *rlp30-2* mutants were dipped for approximately 1.5 min. The T1 generation was selected on 0.2% (v/v) BASTA. For *Nicotiana tabacum* transformation, cells were resuspended in 10 mM

MgCl₂. Leaf pieces were incubated in the bacterial suspension for 3 min, subsequently transferred to MS medium with 2% sucrose, and incubated for 48 h in the dark. Transgenic calli were selected on MS medium with corresponding antibiotics. Transgenic plants selected in sterile culture were transferred to soil and grown in the greenhouse under long day conditions. Protein expression in transgenic lines was verified by immunoblotting using anti-RFP (Abcam) and anti-GFP (Thermo Fisher Scientific) antibodies and response to SCFE1[18].

## Transient protein expression and co-immunoprecipitations

For 35S-promoter-driven transient protein expression in *N. benthamiana or N. tabacum*[19], coding sequences of respective genes were amplified using the primers listed in Supplementary Table 3, cloned into pCR™8/GW/TOPO™ (Thermo Fisher Scientific) and subsequently recombined via Gateway cloning into pGWB5 (C-terminal GFP-tag), pGWB14 (C-terminal HA-tag), or pGWB17 (C-terminal Myc-tag)[41]. Likewise, a pENTR/D-Topo (Thermo Fisher Scientific) clone containing the coding sequence for *NbSOBIR1*[42] was recombined with pGWB14, *p35S::SlSOBIR1-HA* in pGWB14 and constructs for *AtSOBIR1-HA, BAK1-myc* and *RLP23-GFP* have been described previously[19,43].

For SCP$^{Ss}$-binding assays, native SCP$^{Ss}$-Myc (pGWB17) or Cys-to-Ser replacement mutants (Supplementary Table 5) together with RLP30-GFP (pGWB5) from different *Arabidopsis* accessions was transiently co-expressed in *N. benthamiana*. On day 2, leaf material was harvested and protein extracts were subjected to immunoprecipitation using GFP-Trap beads (ChromoTek). 200 mg ground leaf material was subjected to protein extraction and immunoprecipitation using GFP-Trap beads (ChromoTek)[19]. Proteins were detected by immunoblotting with tag-specific antibodies raised against epitope tags GFP (dilution 1:5000), HA (dilution 1:3000) or Myc (dilution 1:5000, all Thermo Fisher Scientific), followed by staining with secondary antibodies coupled to horseradish peroxidase and ECL (Cytiva).

## Plant immune responses

Ethylene production was induced in three leaf pieces floating on 0.5 ml 20 mM MES buffer pH 5.7 and the indicated elicitors[44]. After incubation for 4 h, 1 ml air was drawn from sealed assay tubes and the ethylene content was measured by gas chromatography (GC-14A, Shimadzu). ROS accumulation was measured in one leaf piece per well placed in a 96-well plate (Greiner BioOne) containing 100 µl of a solution with 20 µM luminol derivative L-012 (Wako Pure Chemical Industries) and 20 ng/ml horseradish peroxidase (Applichem)[44]. Luminescence was measured in 2 min intervals both before (background) and for up to 60 min after PAMP or mock treatment using a Mithras LB 940 luminometer (Berthold Technologies). For MAPK activity assays, treated leaf material was harvested after indicated times and frozen in liquid nitrogen. Proteins were extracted with 20 mM Tris-HCl pH 7.5, 150 mM NaCl, 1 mM EDTA, 1% Triton X-100, 0.1% SDS, 5 mM DTT, Complete Protease Inhibitor Mini, EDTA-free (Roche), PhosStop Phosphatase Inhibitor Cocktail (Roche), separated via 10% SDS-PAGE, and transferred to nitrocellulose. Activated MAPKs were detected by protein blotting using the rabbit anti-phospho-p44/42-MAPK antibody (Cell Signalling Technology, 1:1000 dilution)[44]. RT-qPCR analysis in transgenic RLP30-RFP- and AtSOBIR1-GFP-expressing tobacco lines *#49* and *#55* was performed using gene specific primers for *NtPR1* and *NtPDF1.2* and *NtACT9* as house-keeping gene as listed in Supplementary Table 4.

## Plant infections

Leaves of 6–8-week-old *Arabidopsis* plants were primed by leaf infiltration with 1 µM nlp20, 1 µM SCP$^{Ss}$ or SCP-like$^{Psp}$, or water (mock treatment) and after 24 h inoculated with *Pseudomonas syringae* pv. *tomato* DC3000 (*Pst* DC3000) by leaf infiltration with a final cell density of 10⁴ colony-forming units (CFU) per ml 10 mM MgCl₂. Bacterial populations were determined 3 d after infiltration[45]. *N. tabacum* plants were likewise infected with *P. syringae* pv. *tabaci*[46] at a final cell density of 10⁵ CFU/ml and harvested after 4 days. For fungal infection, spores of *B. cinerea* isolate B05.10 or SCP$^{Bc}$ depletion mutant ΔSCP$^{Bc}$/*Bcplp1*[21] (lines 45 and 46) were diluted to a final concentration of 5 ×10⁶ spores per ml in potato dextrose broth and 5-µl drops were used for leaf inoculation. Lesion sizes were determined after 2 d by determining the average lesion diameter. *N. tabacum* leaves inoculated with 0.8 cm fresh mycelial plugs of *Phytophthora capsici* grown on V8 juice agar were clipped and kept in the dark in plastic boxes to obtain high humidity. Lesion diameters were measured at 48 hpi using UV light.

For *Agrobacterium*-mediated virus-induced gene silencing (VIGS)[42], a 300 bp fragment of *REO2* was amplified using primers listed in Supplementary Table 3 and cloned into vector pYL156[47] (*TRV2::REO2*), as control the *TRV2::GUS* construct was used[42]. Gene silencing was verified by measuring *REO2* transcript levels versus *NbActin* transcript levels using quantitative RT-PCR[28] with primers listed in Supplementary Table 4.

## Phylogeny reconstruction

Full-length protein sequences, obtained from Ngou et al.[48] (a) and BLAST searches[49] on Phytozome[50] (b) or Uniprot[51] (c), of *A. thaliana* (a, b), *B. rapa* (a), *M. truncatula* (b), *V.vinifera* (b), *S. lycopersicum* (a, b), *S. tuberosum* (b), *N. benthamiana* (a), *N. sylvestris* (c) and *A. trichopoda* (b) were aligned using MAFFT[52] (--localpair --maxiterate 10000 --reorder). The phylogenetic tree was constructed using IQTree[53] (-T AUTO -m TEST -b 50 -con).

## Statistics and reproducibility

No statistical methods were used to predetermine sample size. The experiments were not randomized. The investigators were not blinded to allocation during experiments and outcome assessment. Data were plotted using GraphPad Prism 9. Data were represented as the mean ± S.D. or as box-and-whisker plots in which the center line indicates the median, the bounds of the box indicate the 25th and 75th percentiles, and the whiskers indicate 1.5 times the interquartile range. Unless otherwise stated, graphs present data from a single experiment. No data were excluded from analyses. Data were analyzed with a two-sided Student's *t* test using Microsoft Office Excel. A summary of statistical analyses is provided in Source data.

## Reporting summary

Further information on research design is available in the Nature Portfolio Reporting Summary linked to this article.

## Data availability

All data are available in the main text or the Supplementary Information; sequence data from this article can be found in Supplementary Table 2. Source Data including detailed corresponding statistics are provided with this paper. The LC-MS/MS data generated in this study as well as the MaxQuant output files have been deposited to the ProteomeXchange Consortium via the PRIDE partner repository (https://www.ebi.ac.uk/pride/) and can be accessed using the identifier PXD036013. Source data are provided with this paper.

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

## Acknowledgements

We thank M. H. A. J. Joosten for *SlSOBIR1* and *NbSOBIR1* constructs and *TRV2::GUS*, Caterina Brancato for assistance in tobacco transformation, and Franziska Hackbarth and Hermine Kienberger for excellent laboratory assistance at the BayBioMS. This work was supported by grants from the Deutsche Forschungsgemeinschaft (DFG) to A.A.G. (SFB766, Gu 1034/3-1), the China Scholarship Council (CSC) to Y.Y. and the BMBF-funded de.NBI Cloud within the German Network for Bioinformatics Infrastructure (de.NBI). We acknowledge support from the Open Access Publication Fund of the University of Tübingen.

## Author contributions

Y.Y., C.E.S., G.F., and A.A.G. conceived and designed the experiments; Y.Y., C.E.S., B.L., C.R., D.K., T.L., and W.Z. conducted experiments; Y.Y., C.E.S., G.F., C.L., M.F.S., A.B., M.H., R.H., and A.A.G. analyzed data; B.G. and H.G. provided materials; G.F. and T.N. revised the manuscript and Y.Y., C.E.S., and A.A.G. wrote the manuscript. All authors discussed the results and commented on the manuscript.

## Funding

## Competing interests

The authors declare no competing interests.
