## [Peer Review File · Nature Communications]

REVIEWER COMMENTS

Reviewer #1 (Remarks to the Author):

In this study, Yang et al. identified a small cysteine-rich protein (SCP) that can be detected by RLP30 of Arabidopsis and RE02 of *N. benthamiana*. Treatment of SCP to the plants induces RLP30/RE02-dependent ethylene production. Interestingly, the modes of sensing SCP are different among Arabidopsis, *N. benthamiana*, and *B. napus*. The SCP identified from different pathogens are recognized by RLP30 and overexpression of RLP30 confers increased resistance to different pathogens. Overall, Yang et al. provided very solid results to support the conclusion that the RLP30/RE02 can perceive the PAMP from three different kingdoms. Although the molecular identity of the SCP-likePSP was not confirmed, they provided very intriguing and novel findings that should be of interest to the community. Please see below for some comments that may help to improve the manuscript.

Major:

1. The term "convergent evolution" is used in the title; however, the only support for this is the low sequence similarity between RLP30 and RE02. A phylogenetic tree including multiple RLP sequences, perhaps similar to the one in Kang et al 2018 (DOI: 10.5423/PPJ.OA.02.2018.0032), should be provided to support that RLP30 and RE02 are with different origins.

Minor:

- 1. In line 97 and a few other lines, "extended data" should be changed to "supplementary Fig" to be consistent.**
- 2. In line 95, re-purified the "activity" of, this should be rephrased to make it reads smoother**
- 3. In Figure 3, point mutation variants "C1S, C2S... etc" should be briefly explained in the figure legend.**

Reviewer #2 (Remarks to the Author):

The overall findings of this study are exciting and of sufficient novelty and impact to readily merit publication in Nature Communications. The manuscript spans impressive breadth. It would be enough for a strong paper to independently discover the small cysteine-rich protein (SCP) ligand for the previously identified pattern recognition receptor RLP30 and discover that RLP30-activating SCP ligands are made by strikingly diverse pathogen taxa (fungi and oomycetes). This manuscript goes well beyond that to also report a second RLP30-activating ligand made by bacteria, providing the striking result that the RLP30 receptor helps plants to sense bacteria, oomycetes and fungi, and reporting that solanaceous plants carry a distinct SCP-sensing receptor that is not orthologous to Brassicaceae RLP30. The study thus describes in one place a holistic and instructive story about plant immune system functional efficiency, conservation and diversification of function. The novelty is diminished in that versions of SCP ligand from other fungi and oomycetes were previously reported (Nie 2019, ref. 12), and the Nicotiana receptor of SCP was also very recently reported (Nie 2022, ref. 22). But the present manuscript brings together a novel story, with multiple entirely new discoveries and a breadth of scope that justifies publication in a high-tier journal.

The methods used are diverse and robust (e.g., strong controls, more than one approach often used to support the same finding), so I have no critique about the experiments that

are presented.

1) There is one additional experiment I would request of the authors, to examine and report if there is constitutive elevation of defense in the RLP30-RFP + AtSOBIR1-GFP N. tabacum transgenic lines #49 and #55, in the absence of pathogens. Simplest would probably be to check mRNA abundance of standard salicylic acid and jasmonic acid response genes PR-1a and PDF1.2 by simple RT-PCR assays. Elevated defense associated with chronic elevation of SA or JA pathways is highly common and alerts researchers to probable indirect/pleiotropic phenomena, rather than straightforward/direct PAMP stimulation of RLP30-mediated defenses.

2) It would also be good to add the word "quantitatively" to the abstract on line 44 ("resulted in quantitatively lower susceptibility") as the word "quantitative" is a recognized term among plant disease resistance researchers that indicates an incremental rather than qualitative shift in resistance, consistent with the results of Fig. 5c and 5d. Similarly on line 280, this should say "confer quantitative resistance".

3) The authors might say a bit more in Concluding Remarks (and optionally, find a place in the abstract if space permits, and the final Introduction paragraph) to highlight the intriguing fact that, despite the broad capacity of multiple Brassicaceae and Solanaceae species to detect SCP-Ss, multiple Arabidopsis ecotypes (rather than just one or two) carry altered RLP30 receptors that do not recognize SCP-Ss. Is this evidence of probable functional diversification?

4) The inability to date to precisely identify the Pseudomonas elicitor of RLP30 seems acceptable, because those sections do offer well-supported and relevant new discoveries (RLP30 also senses a ligand from bacterial pathogens, the ligand is not closely related to fungal/oomycete SCP, and the ligand has been chemically purified to the extent that it is apparently a protein stabilized by Cys-Cys bridges, Solanaceae do not respond to this bacterial SCP-like compound). The paper is stronger with these results than without them.

5) Although RLP30 functioned modestly in Nicotiana, the fact that REE02 from Nicotiana functions in Arabidopsis is also of interest (deserves to be highlighted a bit more if possible, although I agree it doesn't merit mention in abstract if abstract length is limited). It indicates that the partner proteins are sufficiently conserved (cross-functional) across wide taxonomic space – a finding of both basic and possible applied interest. For readers, fit that in with other examples of PAMP perception complex proteins that work across taxa? The concise "Results and Discussion" format of the paper is very successfully used, but perhaps in the vicinity of line 280 the authors could use more words to say a bit more (including a clarification/reminder that Arabidopsis RLP30 didn't work well in Nicotiana without bringing along Arabidopsis SOBIR1).

Line 259 "number" not "amount"

Reviewer #3 (Remarks to the Author):

In this manuscript, the authors identified a small cysteine-rich protein (SCP), which is conserved in many fungi and oomycetes, as a RLP30 ligand. Further study reveals that the Nicotiana benthamiana RLP RE02 also recognizes SCP. Interestingly, RLP30 and RE02 share little sequence similarity and may respond to different parts of SCP. In addition, some Brassicaceae plant species rather than Arabidopsis respond to a linear SCP peptide,

suggesting that distinct plant species are subject to convergent evolution to recognize SCP. Furthermore, the authors claimed that RLP30 shows a ligand specificity for an SCP non-homologous protein secreted by bacterial Pseudomonads. The *N. tabacum* plants expressing RLP30 exhibit enhanced resistance to bacterial, fungal and oomycete pathogens, indicating that Arabidopsis RLP30 is involved in immunogenic patterns detection from three microbial kingdoms. RLP30 may be used as a valuable genetic tool to boost crop resistance. The authors did a lot of work and presented many experimental data. The manuscript is also well written. However, to my knowledge, some important points need to be addressed to make a solid conclusion.

Major concerns:

My major concern is that the authors need to rule out the possibility of contaminations. The authors conclude that SCPSs perception by the RLPs RLP30 and RE02 depends on the SCPSs tertiary structure since SCPSs is sensitive to treatment with DTT (Line 165). This point is difficult to be explained because the heat treatment (95°C) for 1 h doesn't affect SCPSs's elicitation ability at all (S7A). In my opinion, the heat treatment can also destroy protein tertiary structure. According to these data, SCPSs might not be a real elicitor. Is it possible that the elicitor came from the contamination during protein purification? Did the authors attempt to purify SCP from *E. coli*? Protein purification from different sources may explain this. Another possibility is that some SCPSs interacting components or post-translationally modified molecular structure have the eliciting ability? The authors failed to identify SCP-like Psp protein, which provides indirect evidence to support my query.

1: In Fig 1e, the authors present the Co-IP data to support SCPSs interaction with RLP30. However, the interaction might be indirect. Here, if the author could perform in vitro GST pull-down assay, it will be more convincing to make a conclusion.

2. The authors showed that C1S-C8S lost the elicitation ability in Arabidopsis in Fig 3b. The authors should check if the Cys point mutations affect its interaction with RLP30. The results from these experiments might help make a conclusion.

3. How about the interactions between RLP30 from Br-0, Gu-0, Lerik1-3, Sq-0 and SCPSs?

Minor suggestions,

1: Line 125, "Sq-1" should be "Sq-0"? No "Sq-1" accessions are shown in figure S3a.

"Mt-0, Br-0, Lov-1, Lov-5, ICE111 and Sq-1" should be "Mt-0, Bak-2, Lov-1, Lov-5, ICE111"? Which matches the data shown in figure S3b.

2: Line 130 and figure S4a, the "Sq-1" should be changed to "Sq-0" ? Only "Sq-0" is shown in figure S3a, not sure which one is correct.

Figure S4a, are RLP30 from Bak-2, Gu-0, and Lerik1-3 also have similar amino acid substitutions?

3: Line 151, Fig. 1e should be Fig. 1d?

SCPSs recognition system is present in *N. benthamiana* (Fig. 1d), and homologs from *B. cinerea* (SCPBc)

Other recommendations in writing :

1: Line 43, change "a SCP-nonhomologous" to "an SCP-nonhomologous"

2: Line 146, remove "in" before "SCPSs-insensitive accessions"

3: Line 407, change "35" to "35S"

Reply to comments of Reviewer 1

Major issues

Q1: The term “convergent evolution” is used in the title; however, the only support for this is the low sequence similarity between RLP30 and RE02. A phylogenetic tree including multiple RLP sequences, perhaps similar to the one in Kang et al 2018 (DOI: 10.5423/PPJ.OA.02.2018.0032), should be provided to support that RLP30 and RE02 are with different origins.

R1: We thank Referee 1 for her/his valuable recommendation and now provide a phylogenetic tree including multiple RLP sequences as shown in new Supplementary Fig. 7. Phylogenetic analysis supported our claim that RLP30 and RE02 are derived from convergent evolution. Along with Supplementary Fig. 7, we added a new paragraph on RLP30/RE02 evolution entitled “RLP30 and RE02 evolved independently for at least 140 million years” (page 6, lines 173-195).

Moreover, we added in the conclusion a paragraph on RLP30 and RE02 differences, this paragraph now reads as follows (page 11, lines 313-321):

Convergent evolution of RLP30 and RE02 is supported by our phylogenetic analysis, indicating that RLP30 and RE02 subfamilies evolved independently for at least 140 million years. Moreover, RE02 and RLP30 have different protein structures with RE02 harboring 28 LRR motifs versus 21 LRRs in RLP30 and both proteins have highly divergent island domains. Based on the motif features within the island domain, RE02 (Kx5Y motif) and RLP30 (Hx8KG motif) are classified into two different clades²⁶. Notably, the island domain of plant LRR-proteins was shown to participate in ligand binding^{15,27}, explaining the different structural requirements of RLP30 and RE02 for SCPSs perception and the ability of RLP30 to recognize a bacterial SCP-like pattern.

Minor issues

Q2: In line 97 and a few other lines, “extended data” should be changed to “supplementary Fig” to be consistent.

Q3: In line 95, re-purified the “activity” of, this should be rephrased to make it reads smoother

Q4: In Figure 3, point mutation variants “C1S, C2S... etc” should be briefly explained in the figure legend.

R2 to R4: All changes are made as suggested by Referee 1.

Reply to comments of Reviewer 2

Major issues

Q1: There is one additional experiment I would request of the authors, to examine and report if there is constitutive elevation of defense in the RLP30-RFP + AtSOBIR1-GFP *N. tabacum* transgenic lines #49 and #55, in the absence of pathogens. Simplest would probably be to check mRNA abundance of standard salicylic acid and jasmonic acid response genes PR-1a and PDF1.2 by simple RT-PCR assays. Elevated defense associated with chronic elevation of SA or JA pathways is highly common and alerts researchers to probable indirect/pleiotropic phenomena, rather than straightforward/direct PAMP stimulation of RLP30-mediated defenses.

R1: We fully agree with Reviewer 2 who requested to examine whether transgenic *RLP30/AtSOBIR1*-transgenic *N. tabacum* lines exhibit constitutive elevation of defense. As suggested, we conducted RT-qPCR analysis on *N. tabacum* transgenic lines #49 and #55 using gene specific primers for the salicylic acid marker gene *PR-1a* (according to Riviere et al., 2008, <https://doi.org/10.1093/jxb/ern044>) and the jasmonic acid marker gene *PDF1.2* (according to Jin et al., 2017, <https://doi.org/10.3389/fpls.2017.02263>). Both *PR-1* and *PDF1.2* transcript levels were not elevated in non-stimulated transgenic tobacco lines compared to the wild-type control. In addition, we neither

observed elevated ethylene levels in non-stimulated or mock-treated leaves of the transgenic *RLP30/AtSOBIR1*-transgenic *N. tabacum* lines (see Fig. 5a) nor in tobacco leaves transiently expressing these proteins alone or in combination (see Supplementary Figure 10a). These results suggest that ectopic expression of Arabidopsis *RLP30* and *SOBIR1* in tobacco does not lead to constitutive elevation of defense responses in the absence of the SCP stimulus.

We added the results of the RT-qPCR analysis as Supplementary Figure 10d and refer to these results in the main text accordingly (page 9, lines 285-289):

Neither of the two transgenic lines exhibited autoimmune phenotypes as mock treatment did not induce ethylene accumulation and no elevated expression of salicylic acid marker gene *PR-1a* or jasmonic acid marker gene *PDF1.2* could be detected in untreated plants (Fig. 5a and Supplementary Fig. 10a,d).

Minor issues

Q2: It would also be good to add the word “quantitatively” to the abstract on line 44 (“resulted in quantitatively lower susceptibility”) as the word “quantitative” is a recognized term among plant disease resistance researchers that indicates an incremental rather than qualitative shift in resistance, consistent with the results of Fig. 5c and 5d. Similarly on line 280, this should say “confer quantitative resistance”.

R2: These changes were made as suggested.

Q3: The authors might say a bit more in Concluding Remarks (and optionally, find a place in the abstract if space permits, and the final Introduction paragraph) to highlight the intriguing fact that, despite the broad capacity of multiple Brassicaceae and Solanaceae species to detect SCP-Ss, multiple Arabidopsis ecotypes (rather than just one or two) carry altered RLP30 receptors that do not recognize SCP-Ss. Is this evidence of probable functional diversification?

R3: We thank this Reviewer for this suggestion and added the following to the concluding remarks (page 11, lines 322-330):

Intriguingly, despite the broad capacity of multiple Brassicaceae and Solanaceae species to detect SCPs, multiple *Arabidopsis* ecotypes carry altered RLP30 receptors that do not recognize SCPs. Although RLP30 is considered a conserved LRR-RP and RLP30 sequences are present in most *Arabidopsis* ecotypes²⁸, slight sequence variations might lead to functional diversification. Alternatively, accession-specific loss of RLP30 function might be explained by redundancy of *Arabidopsis* cell-surface immune receptors for fungal patterns such as SCPs, NLPs, PGs, oligogalacturonides or chitin perceived by *Arabidopsis* receptors RLP30, RLP23, RLP42, WAK1 or LYK5/CERK1, respectively^{19,29-31}. This complexity is also reminiscent of *Arabidopsis* sensors for *Pseudomonas*-derived patterns⁴.

Q4: The inability to date to precisely identify the *Pseudomonas* elicitor of RLP30 seems acceptable, because those sections do offer well-supported and relevant new discoveries (RLP30 also senses a ligand from bacterial pathogens, the ligand is not closely related to fungal/oomycete SCP, and the ligand has been chemically purified to the extent that it is apparently a protein stabilized by Cys-Cys bridges, Solanaceae do not respond to this bacterial SCP-like compound). The paper is stronger with these results than without them.

R4: We acknowledge and share Reviewer 2's point of view that, even though the molecular identity of the *Pseudomonas* activity remains elusive to date, these results support and highlight the fact that *Arabidopsis* RLP30 is a unique pattern recognition receptor that recognizes unrelated patterns from three microbial kingdoms.

Q5: Although RLP30 functioned modestly in Nicotiana, the fact that RE02 from Nicotiana functions in Arabidopsis is also of interest (deserves to be highlighted a bit more if possible, although I agree it doesn't merit mention in abstract if abstract length is limited). It indicates that the partner proteins are sufficiently conserved (cross-functional) across wide taxonomic space – a finding of both basic and

possible applied interest. For readers, fit that in with other examples of PAMP perception complex proteins that work across taxa? The concise “Results and Discussion” format of the paper is very successfully used, but perhaps in the vicinity of line 280 the authors could use more words to say a bit more (including a clarification/reminder that Arabidopsis RLP30 didn’t work well in Nicotiana without bringing along Arabidopsis SOBIR1).

R5: We thank this Reviewer for this suggestion and added the following to the conclusion section (page 12, lines 336-343):

Notably, RE02 also functions in *Arabidopsis*, indicating that the complex partner proteins are sufficiently conserved across wide taxonomic space, which is also supported by multiple reports on successful transfer of PRRs across taxa^{14,19,32,33}. As RLP30, however, only modestly functioned in tobacco without co-expressing AtSOBIR1, we propose that co-expression of heterologous PRR/co-receptor pairs may be deployed to overcome putative incompatibilities of ectopically expressed PRRs. Consequently, this way enhanced broad-spectrum resistance may be engineered as compared to expressing single PRRs alone.

Q6: Line 259 “number” not “amount”

R6: This change was made as suggested.

Reply to comments of Reviewer 3

Major issues

Q1: My major concern is that the authors need to rule out the possibility of contaminations. The authors conclude that SCP^{Ss} perception by the RLPs RLP30 and RE02 depends on the SCP^{Ss} tertiary structure since SCP^{Ss} is sensitive to treatment with DTT (Line 165). This point is difficult to be explained because the heat treatment (95°C) for 1 h doesn’t affect SCP^{Ss}’s elicitation ability at all (S7A). In my opinion, the heat treatment can also destroy protein tertiary structure. According to these data, SCP^{Ss} might not be a real elicitor. Is it possible that the elicitor came from the contamination during protein purification? Did the authors attempt to purify SCP from *E. coli*? Protein purification from different sources may explain this. Another possibility is that some SCP^{Ss} interacting components or post-translationally modified molecular structure have the eliciting ability? The authors failed to identify SCP-like Psp protein, which provides indirect evidence to support my query.

R1: We agree with the Reviewer that heat treatment can destroy tertiary structures by disrupting the alpha-helices and beta-sheets in a protein which then uncoils into a random shape. Thus, heat treatment can be used to disrupt hydrogen bonds and non-polar hydrophobic interactions. However, some proteins are intrinsically heat-stable due to their structure, which can be further stabilised by covalent bonds such as disulfide bridges. Here, heat is less efficient in disrupting this bond. Rather, reducing agents such as DTT are used to denature disulfide bonds. As SCP^{Ss} has multiple predicted disulfide bridges (see Fig. 3a) and as, particularly in *Arabidopsis*, the involved cysteines all play a crucial role for immunogenic activity (see Fig. 3b), it is likely that SCP^{Ss} is resistant to heat treatment, similar to what we showed for the SCP^{Ss}-containing crude SCFE1 preparation (Supplemental Figure 1D in Zhang *et al.*, Plant Cell 2013).

We also agree with the Reviewer that potential contaminations during protein purification might occur. We never produced fungal SCP^{Ss} in *E. coli* as a prokaryotic expression system. However, to rule out that these contaminations or any SCP-copurifying protein are responsible for the RLP30-dependent defense response we present in the manuscript two unrelated eukaryotic expression systems: i) expression in the *Nicotiana benthamiana* apoplast (see Supplementary Fig.1f) and ii) expression in *Pichia pastoris* (see Fig. 1a). SCP^{Ss} from both sources induced RLP30-dependent ethylene production. Moreover, the preparations of SCP truncations C1C5, C3C8 and C1C4 produced in *Pichia* were not active in *Arabidopsis*, ruling out co-purification of SCP-unrelated elicitor-activities during the purification

procedure. Finally, the C3C5 fragment of SCP^{Ss} was synthetic, yet presenting a third independent source for elicitor-activity. In sum, we present in our manuscript three different sources for SCP^{Ss} or related fragments, all of which displayed elicitor activity in at least one of the tested plant species (*Arabidopsis*, *Nicotiana*, *Brassica*), hence making it highly unlikely that a co-purified contamination is responsible for the observed defense responses.

In our opinion, finding SCP-like activities in bacteria (that are, however, NOT recognized in Solanaceae) rather strengthens our claim that SCP^{Ss} and bacterial SCP-like structures, and not co-purified contaminations, have immunogenic activity as the bacterial system can be considered a fourth independent source.

To acknowledge this Reviewers concerns, we made the following addition to the results section (underlined below, page 4, lines 106-111):

To confirm SCP^{Ss} as the ligand of RLP30, the SCP^{Ss} protein was heterologously expressed in *N. benthamiana* or *Pichia pastoris* (Supplementary Fig. 1d,e), with C-terminal GFP or His tags, respectively. Both forms were found to induce ethylene production in *Arabidopsis* Col-0 wild-type plants but not in the *rlp30-2* mutant (Fig. 1a and Supplementary Fig. 1f), ruling out that contaminations occurring during the purification procedure or SCP^{Ss}-copurifying proteins are responsible for the RLP30-dependent defense response.

Q2: In Fig 1e, the authors present the Co-IP data to support SCP^{Ss} interaction with RLP30. However, the interaction might be indirect. Here, if the author could perform *in vitro* GST pull-down assay, it will be more convincing to make a conclusion.

R2: We agree with this Reviewer that Co-IP analyses do not proof direct interaction of a ligand and its receptor. However, this assay was used in several studies to show association of RLPs and their ligands such as INF1-REL interaction (Chen *et al.*, Plant Cell 2023) or XEG1-RXEG1 interaction (Wang *et al.*, Nature Commun. 2018). Hence, to date this assay can be considered state-of-the-art. Nevertheless, as suggested by this Reviewer, we expressed RLP30-LRRs fused to the maltose-binding protein (MBP) in *E. coli*. The MBP-fusion protein could be expressed, however, not as soluble protein. As RLP30 is highly glycosylated (as can be deduced from the increased size in Western Blot analysis such as Fig. 1e) we next expressed RLP30-LRRs in insect cells. Again, the fusion protein was detectable in cell pellets and not the soluble protein fraction. We therefore were not able to purify any functional RLP30-LRRs from *E. coli* or insect cells to perform any *in vitro* pull-down assays.

Finally, the expression of the LRR-domain of RLPs, in contrast to LRR-RLKs, generally seems to be difficult and so far, the only crystal structure available for an LRR-RLP is the one of the LRR-RLP RXEG1 from *Nicotiana benthamiana* that recognizes XEG1 xyloglucanase from the pathogen *Phytophthora sojae* (Sun *et al.*, Nature 2022).

To acknowledge this Reviewers concerns we replaced the term “interaction” by “complex formation” as follows (underlined):

Page 4, lines 122/123:

“To assess whether SCP^{Ss} can be found in complex with RLP30, co-immunoprecipitation assays were performed with extracts from *N. benthamiana* leaves that...”

Page 5, lines 134-139:

“Thus, the single amino acid substitutions in different parts of the LRR and island domains, which are specific to SCP^{Ss}-responsive RLP30 variants¹⁸, such as L307R in Mt-0, R433G in Lov-1 and Lov-5, R433L in ICE111, N561Y in Bak-2, G563V in Br-0, and F760 in Gu-0 (Supplementary Fig. 4), respectively, all affect RLP30/SCP^{Ss} complex formation and, likely, are responsible for the non-functionality of these mutated receptors.

Q3: The authors showed that C1S-C8S lost the elicitation ability in Arabidopsis in Fig 3b. The authors should check if the Cys point mutations affect its interaction with RLP30. The results from these experiments might help make a conclusion.

R3: As suggested by Reviewer 3, we now also performed co-immunoprecipitation experiments with C to S replacement mutants of SCP^{Ss}. As shown in Supplemental Figure 8c, none of the mutated, myc-tagged SCP^{Ss} versions could be co-purified by RLP30-GFP.

Q4: How about the interactions between RLP30 from Br-0, Gu-0, Lerik1-3, Sq-0 and SCP^{Ss}?

R4: We also performed these suggested co-immunoprecipitation experiments with SCP^{Ss} and RLP30 versions from Br-0, Gu-0, Lerik1-3, Sq-1 and could not see any complex formation of these RLP30 versions compared to RLP30 from the Col-0 accession. Notably, RLP30 from Sq-1 and Lerik1-3 could not be transiently expressed in *N. benthamiana*. These results are now included in Supplementary Fig. 3b.

Minor issues

Q5: Line 125, “Sq-1” should be “Sq-0”? No “Sq-1” accessions are shown in figure S3a. “Mt-0, Br-0, Lov-1, Lov-5, ICE111 and Sq-1” should be “Mt-0, Bak-2, Lov-1, Lov-5, ICE111”? Which matches the data shown in figure S3b.

R5: All Sq-0 labelling (in all figures, legends, text) was changed to Sq-1, as this is the correct name of this accession. Also, the enumeration “Mt-0, Br-0, Lov-1, Lov-5, ICE111 and Sq-1” was replaced by “these insensitive accessions” (page 5, line 131/132).

Q6: Line 130 and figure S4a, the “Sq-1” should be changed to “Sq-0” ? Only “Sq-0” is shown in figure S3a, not sure which one is correct.

Figure S4a, are RLP30 from Bak-2, Gu-0, and Lerik1-3 also have similar amino acid substitutions?

R6: We apologize for this mistake: all Sq-0 labelling (in all figures, legends, text) was changed to Sq-1, as this is the correct name of this accession, as also mentioned above for Q4.

We also now include all amino acid substitutions in Supplementary Fig. 4a and provide all RLP30 nucleotide sequences in the source data file.

Q7: Line 151, Fig. 1e should be Fig. 1d?

Q8: Line 43, change “a SCP-nonhomologous” to “an SCP-nonhomologous”

Q9: Line 146, remove “in” before “SCPSs-insensitive accessions”

Q10: Line 407, change “35” to “35S”

R7 to R10: All corrections are made as suggested by the reviewer.

REVIEWERS' COMMENTS

Reviewer #1 (Remarks to the Author):

The authors have sufficiently addressed the comments I raised last time. I think this manuscript is ready to be published.

Reviewer #3 (Remarks to the Author):

In this much improved version of the manuscript, the author addressed all my comments and concerns so I will gladly support this manuscript for publication.

Reply to comments of Reviewers

The Reviewers had no further requests, see below.

Reviewer #1 (Remarks to the Author):

The authors have sufficiently addressed the comments I raised last time. I think this manuscript is ready to be published.

Reviewer #3 (Remarks to the Author):

In this much improved version of the manuscript, the author addressed all my comments and concerns so I will gladly support this manuscript for publication.